# Assessing criticality in pre-seizure single-neuron activity of human epileptic cortex

**Annika Hagemann** [1], **Jens Wilting** [1], **Bita Samimizad** [2], **Florian Mormann** [2], **Viola Priesemann** [1,3] *

1 Max-Planck-Institute for Dynamics and Self-Organization, Göttingen, Germany, 2 Department of Epileptology, University of Bonn Medical Centre, Bonn, Germany, 3 Bernstein Center for Computational Neuroscience (BCCN) Göttingen, Germany

* viola.priesemann@ds.mpg.de

**Data Availability Statement:** The binned activity time series from full MTL, hippocampus, entorhinal cortex, parahippocampal cortex and amygdala from all 20 patients are available on GIN (G-Node

## Abstract

Epileptic seizures are characterized by abnormal and excessive neural activity, where cortical network dynamics seem to become unstable. However, most of the time, during seizure-free periods, cortex of epilepsy patients shows perfectly stable dynamics. This raises the question of how recurring instability can arise in the light of this stable default state. In this work, we examine two potential scenarios of seizure generation: (i) epileptic cortical areas might generally operate closer to instability, which would make epilepsy patients generally more susceptible to seizures, or (ii) epileptic cortical areas might drift systematically towards instability before seizure onset. We analyzed single-unit spike recordings from both the epileptogenic (focal) and the nonfocal cortical hemispheres of 20 epilepsy patients. We quantified the distance to instability in the framework of criticality, using a novel estimator, which enables an unbiased inference from a small set of recorded neurons. Surprisingly, we found no evidence for either scenario: Neither did focal areas generally operate closer to instability, nor were seizures preceded by a drift towards instability. In fact, our results from both pre-seizure and seizure-free intervals suggest that despite epilepsy, human cortex operates in the stable, slightly subcritical regime, just like cortex of other healthy mammalians.

## Author summary

In epilepsy patients, the brain regularly fails to control its activity, resulting in epileptic seizures. So far, it is not fully understood why the brains of epilepsy patients are susceptible to seizures and what the mechanism behind seizure generation is. We investigated epilepsy from the perspective of collective neural dynamics in the brain. It has been hypothesized that epileptic seizures might be a tipping over from stable, so-called subcritical, dynamics (which are commonly found in healthy brains) to unstable, so-called supercritical dynamics. We therefore examined two potential scenarios of seizure generation: (i) epileptic brain areas might generally operate closer to instability, which would make epilepsy patients generally susceptible to seizures, or (ii) epileptic brain areas might slowly drift towards instability before seizure onset. To test these two hypotheses, we analyzed activity of single neurons recorded with micro-electrodes in epilepsy patients. Contrary to

Infrastructure), repository: https://gin.g-node.org/annika.hagemann/binned_spiking_activity_human_epilepsy.

**Funding:** F.M. acknowledges support from the Volkswagen Foundation, the German Ministry of Education and Research (BMBF 031L0197B) and the German Research Council (DFG MO 930/8-1, SFB 1089). J.W. received support from the Gertrud-Reemstma-Stiftung. J.W and V.P. received financial support from the German Ministry for Education and Research (BMBF) via the Bernstein Center for Computational Neuroscience (BCCN) Göttingen. A.H., J.W. and V.P. received financial support from the Max Planck Society. The funders had no role in study design, data collection and analysis, decision to publish, or preparation of the manuscript.

**Competing interests:** The authors have declared that no competing interests exist.

widespread expectation, we found no evidence for either scenario, thus no evidence that epilepsy involves a transition to supercritical collective neural dynamics. In fact, our results from both seizure-free and pre-seizure intervals suggest that the human epileptic brain operates in the stable regime, just like the brains of other healthy mammalians.

## Introduction

The existence of epileptic seizures suggests that cortical networks self-organize to a state that is prone to instability. Interestingly, from a computational perspective, a working point at the border to instability has advantages for information processing, because it renders a network highly sensitive to input and fosters non-stereotypical responses to stimuli [1–4]. It is therefore hypothesized that cortical networks have to strike a balance between maximizing their sensitivity and variability, while maintaining a safety margin to instability [5, 6]. Indeed, evidence is accumulating that cortical networks *in vitro* as well as in healthy, awake animals operate close to a *critical point* which marks a transition between stable (subcritical) and unstable (supercritical) dynamics [1, 2, 6–23].

Epileptic seizures have been hypothesized to reflect a transition to supercritical, unstable dynamics [24–31], thus presenting a failure in keeping a safety-margin from instability. Multiple computational models suggest that entering a seizure could correspond to a change in the brain's dynamical state, via a bifurcation or critical transition [31–33]. Furthermore, experimental evidence suggests that neural activity during epileptic seizures and epileptiform activity deviates from healthy activity in indicators of criticality [26, 27], and neural activity during seizure termination has shown signatures of a critical transition across different recording levels [34]. Recently, long-term changes in signatures of critical slowing down, together with epileptiform discharges, were shown to be a reliable indicator for seizure risk [35]. In contrast to these findings, recent studies on human iEEG found no consistent warning signals of a critical transition prior to seizures [36, 37], raising the question of whether, and under which conditions, the framework of criticality can capture the mechanisms that underlie seizure generation.

Assuming that seizure onset does indeed correspond to loosing the safety margin to supercriticality, it remains open whether the safety margin is in general smaller in certain areas of epileptic brains, or whether the safety margin is lost prior to seizure onset [32]. In the first scenario, one expects brain areas, in which seizures emerge, to generally operate closer to instability than the same areas in a healthy human brain, so that small fluctuations can push the system into a seizure (Fig 1b). In the second scenario, one expects that the distance to criticality diminishes systematically before seizure onset, making the onset predictable (Fig 1c).

We investigated these two hypotheses using extracellular spike recordings from patients with focal epilepsy. In particular, we addressed two questions: (i) Do the brain regions in which the seizures emerge generally operate closer to criticality? (ii) Does the distance to criticality change systematically prior to seizure onset? To that end, we assessed the distance to criticality based on the branching parameter $m$ (Fig 1a). The branching parameter $m$ characterizes the spreading of neural activity. If $m$ is smaller than one, one action potential (spike) on average triggers less than one action potential in the subsequent time step, and the neural network is stable (subcritical regime); if $m$ is larger than one, runaway activity may emerge and the system is unstable (supercritical regime), and $m = 1$ marks the transition between the two (critical state) [38, 39].

                                   

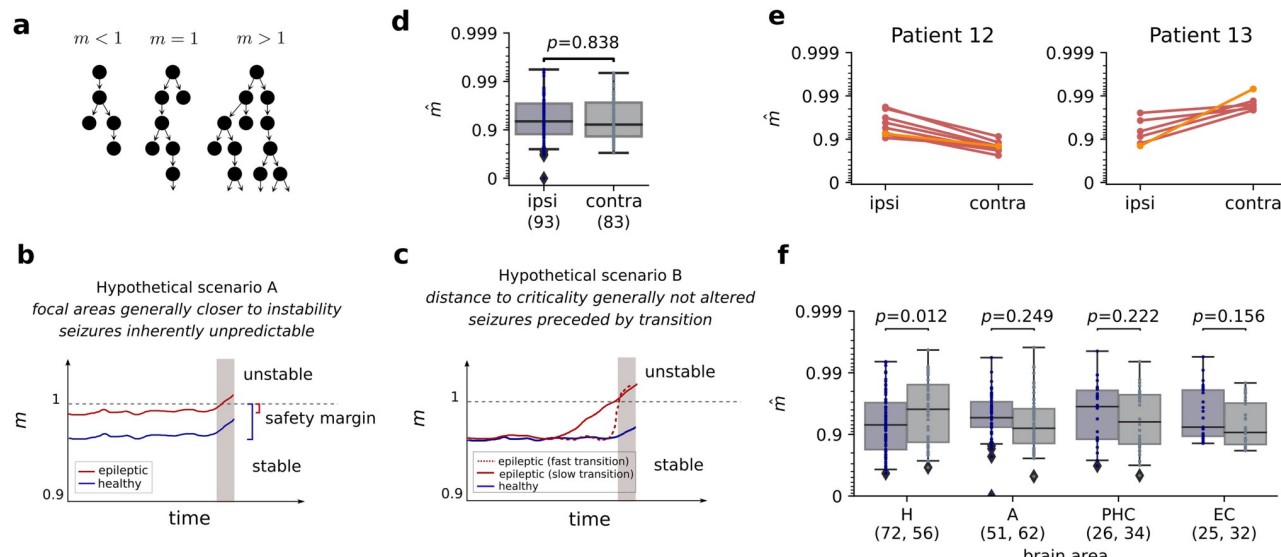

**Fig 1. Comparison of the distance to criticality between the hemisphere containing the epileptic focus and the contralateral hemisphere. a** Branching process approximation of activity propagation in the brain. Both excitatory and inhibitory connections are summarized to an *effective* excitation. Depending on the branching parameter *m*, dynamics are stable ($m < 1$), unstable ($m > 1$) or critical ($m = 1$). **b, c** Hypothetical scenarios of seizure generation. The epileptic focus might, in general, operate closer to criticality and consequently enter the supercritical regime in an unpredictable manner because of small fluctuations (scenario A). Alternatively, dynamics might systematically drift towards instability before seizure onset. If that drift is sufficiently slow, seizures might be predictable (scenario B). **d** The branching parameter $\hat{m}$ of neuronal activity in MTL across recordings and patients shows no significant difference between ipsilateral and contralateral hemisphere (mixed effect ANOVA with patientID as random effect). **e** Estimated $\hat{m}$ in ipsilateral and contralateral hemisphere for multiple recordings of two exemplary patients. While several patients showed a consistent difference between hemispheres across recordings, this difference is not predicted by the location of the epileptic focus (orange: reference recording, red: pre-seizure recording, see S2 Fig for all patients). **f** Comparison of the branching parameter $\hat{m}$ in ipsilateral and contralateral hemisphere for different subregions of the MTL, (hippocampus (H), amygdala (A), parahippocampal cortex (PHC) and entorhinal cortex (EC)). The hippocampus shows a just significant difference between ipsilateral and contralateral hemisphere, with a smaller *m* in the ipsilateral hemisphere. The other subregions show no significant difference ($p < 0.05/4$ required after Bonferroni correction). Box plots show median and quartiles, while whiskers extend to the rest of the distribution, except for outliers (points beyond 1.5 x interquartile range).

By characterizing activity spread, the branching parameter incorporates multiple network properties, whose alterations may be associated with epilepsy, including the excitability of neurons, the excitation-inhibition ratio, and the connectivity [40, 41]. More generally, an increasing branching parameter indicates more temporally correlated neural activity [12] and is therefore a signature of *critical slowing*, which is expected when a system approaches a critical transition [25].

To quantify *m*, we made use of a novel, unbiased estimator that only requires knowing the number of spikes sampled from a small set of neurons to return a reliable estimate of the branching parameter $\hat{m}$ [39]. Importantly, the estimator is invariant under subsampling [42], thus it can infer the propagation of activity in a network even when recording only a small fraction of all neurons [12, 43].

To assess whether epileptic areas generally operate closer to instability, we compared the medial temporal lobe (MTL) containing the epileptic focus to the same area in the contralateral hemisphere, as an approximation of a healthy control. To assess whether the distance to criticality diminishes systematically before seizure onset, we monitored *m* during the last 10 minutes before seizure onset.

## Results

We estimated the distance to criticality in human MTL based on single unit activity from $n = 20$ patients with focal epilepsy. A precise quantification of the distance to criticality has

                                   

become possible with a novel, unbiased estimator [39]. The estimator had returned a branching parameter of $m \approx 0.98$ for various different mammalian species, which corresponds to stable, slightly subcritical dynamics [12, 39]. We applied the same estimator to single unit activity in human MTL, both from the hemisphere containing the epileptic focus (ipsilateral hemisphere), and the contralateral one.

Consistent with recent studies in awake mammalians [5, 6, 12, 13, 44], we found single unit activity in human MTL to reflect dynamics close to criticality, but with a small safety margin to instability (see Fig 1). The recorded activity $A_t$ was vastly consistent with a branching process, as most recordings clearly showed the exponential decay that is expected for the autocorrelation functions (83 out of 91 recordings in the contralateral hemisphere, 93 out of 105 recordings in the ipsilateral hemisphere, see S1 Fig). Across subjects and recordings, the branching parameter indicated dynamics in a slightly subcritical regime, with most recordings being close to criticality ($0.9 < m < 1$, see Fig 1). Hence, our results for human MTL are consistent with those in other mammalian species, suggesting that mammalian cortex in general self-organizes to a slightly subcritical regime.

We tested the hypothesis that the ipsilateral hemisphere generally operates closer to the critical point, which would make it prone to tipping over to unstable dynamics (Fig 1b). However, we found no significant difference between ipsilateral and contralateral hemisphere for $\hat{m}$ across recordings and patients (mixed effect ANOVA with patientID as random effect, $p \approx 0.84$). *Within* some of the patients, however, there were consistent differences in the distance to criticality between hemispheres (Fig 1e and S2 Fig). Out of the 8 patients for whom there were sufficient recordings for a significance test, we found $\hat{m}_{ipsi} < \hat{m}_{contra}$ in 2 patients, $\hat{m}_{ipsi} > \hat{m}_{contra}$ in 2 patients and no significant difference in the remaining 4 patients. Our result thus suggests that there can be consistent differences in the distance to criticality between hemispheres but that the location of the epileptic focus does not predict that difference consistently across patients.

To test whether the distance to criticality in the ipsilateral hemisphere is only altered in specific subregions of the MTL, we analyzed the recorded activity $A_t$ separately in each of the subregions, including amygdala, entorhinal cortex, parahippocampal cortex and hippocampus. Consistent with the above results, we found that single unit activity in all subregions reflects a subcritical regime (Fig 1f). To test for differences between ipsilateral and contralateral hemisphere, we jointly took into account the effects of brain area, location of the epileptic focus and patient-ID. We found a significant interaction effect of brain area and focus (mixed effect ANOVA with patientID as random effect, $p \approx 0.021$). The post-hoc comparisons of the distance to criticality within each subregion revealed a significant difference between ipsilateral and contralateral hemisphere in hippocampus (mixed effect ANOVA with patientID as random effect, $p \approx 0.012$). However, in contrast to our expectation, the results indicate that the hippocampus of the ipsilateral hemisphere operated further away from criticality than the contralateral hippocampus. In all other subregions, we found no significant differences between ipsilateral and contralateral hemisphere (Fig 1f).

The autocorrelation structure of neuronal activity, and therefore the branching parameter $m$, can change depending on vigilance state and circadian rhythm [35, 37, 45, 46]. Furthermore, anti-epileptic drugs modulate cortical excitability and were shown to affect the branching parameter [47, 48]. To make sure that temporal differences do not overshadow differences between the ipsilateral and the contralateral hemisphere, or wrongfully give rise to apparent differences, we performed the same analyses using only *paired* recordings (recordings obtained from the same patients during the same time intervals). Consistent with the results shown in Fig 1d and 1f, we found a significant difference between the ipsilateral and the contralateral

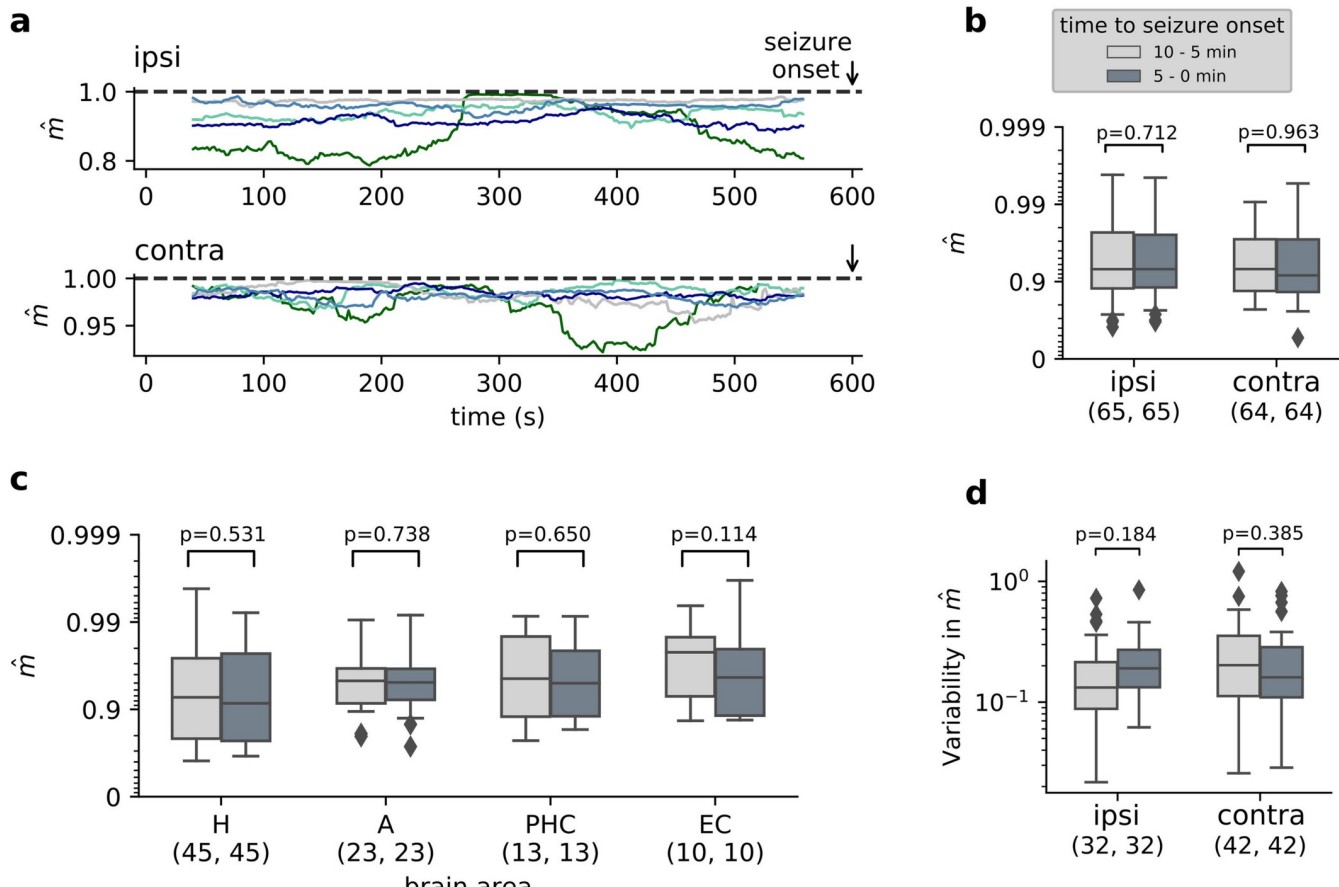

**Fig 2. No consistent change in the distance to criticality before seizure onset. a** Time-resolved estimation of $\hat{m}$ for the full MTL within the last 10 min prior to seizure onset showed variability, but no consistent behavior during the last minutes before seizure onset. Plots show example traces of multiple seizures from one patient. Each trace corresponds to one pre-seizure period, shown both for the ipsilateral and the contralateral hemisphere. Traces of all patients and seizures shown in S6 Fig. **b** No significant difference in the distance to criticality between the last 5 min prior to seizure onset and the previous 5 min—neither in the MTL containing the epileptic focus, nor in the contralateral one (p-values of Wilcoxon signed rank test). **c** Separate analysis of the individual subregions of MTL. Within none of the subregions there were significant differences between the last 5 min before seizure onset and the previous 5 min. Box plots in **c** show results from the hemisphere containing the epileptic focus. **d** Variability of the time-resolved branching parameter $\hat{m}(t)$ in the last 5 min prior to seizure onset and the previous 5 min. Variability is quantified using the median absolute deviation from the median logarithmic distance to criticality (p-values of Wilcoxon signed rank test).

hemisphere in hippocampus, and no significant differences in the remaining subregions and the full MTL (S3 Fig).

Previous studies have reported changes in different characteristics of neuronal activity from seconds up to a time scale of days prior to seizure onset [30, 35, 49, 50]. To investigate whether epileptic seizures are caused by systematically loosing the safety margin to supercriticality (scenario B, Fig 1c), we estimated the branching parameter $m(t)$ in a time-resolved manner during pre-seizure recordings (Fig 2a, see also S6 Fig for all patients). While the branching parameter $\hat{m}$ showed variations over time, none of the patients consistently showed a systematic trend prior to seizure onset.

As a more coarse measure, we compared the branching parameter of the first half to the second half of pre-seizure recordings (Fig 2b). Again, we found no significant difference when approaching seizure onset, neither in the ipsilateral, nor in the contralateral hemisphere (Wilcoxon signed rank test on the logarithmic distance to criticality, $p \approx 0.71$, and $p \approx 0.96$, respectively). Finally, we applied the same analysis to the individual subregions of MTL. None of the

individual subregions showed a significant difference between the first and the second half of pre-seizure recordings (Fig 2c). The results were consistent when additionally accounting for the patientID as a random effect. Thus, across patients and subregions of MTL, we found no evidence for dynamics approaching supercriticality during the last 10 minutes before seizure onset.

Instead of a systematic trend in the distance to criticality before seizure onset, it is conceivable that the variability in the distance to criticality increases. If the variability is high compared to the safety margin, dynamics could enter the supercritical regime spontaneously, driven random fluctuations. To investigate whether fluctuations in the distance to criticality increase when approaching seizure onset, we compared the variability in the time-resolved distance to criticality in the first half and the second half of pre-seizure recordings (Fig 2d). However, we found no significant change when approaching seizure onset across patients and recordings.

So far, we have used the *electrographic* focus for our analyses, i.e. the brain area in which seizures emerged, determined by clinicians prior to surgery. However, surgery outcomes of the patients differ (see S1 Table). We therefore ran the main analyses again, including only patients that were seizure-free after the surgery (Engel scale 1a, 10 out of 20 patients). Consistent with our previous findings, we found no significant change before seizure onset in these patients (S8 Fig). Furthermore, we found no significant differences between ipsilateral and contralateral hemispheres neither in the full MTL, nor in the individual subregions (S7 Fig). In particular, the previously observed increased distance to criticality in ipsilateral hippocampus was not present in this reduced dataset. Note, however, that the number of recordings was smaller compared to the full dataset, such that the required effect sizes for significant results were correspondingly larger.

## Discussion

We started off with the hypotheses that a brain area affected by epilepsy might (i) generally operate closer to criticality (and instability), or (ii) systematically move from the stable, sub-critical to the unstable, supercritical regime before seizure onset. However, we found no evidence for either hypothesis when evaluating single unit activity of human medial temporal lobe. Instead, we found that both hemispheres, the one containing the epileptic focus, and the contralateral one operate in a slightly subcritical regime. Thereby our results in human medial temporal lobe are in line with those on single unit activity from multiple other mammalian species that all show reverberating, slightly subcritical dynamics [12].

We found no evidence that the hemisphere, in which seizures emerge, generally operates closer to instability. Although there were systematic differences between ipsilateral and contralateral hemisphere in some patients, these differences were not consistent across patients. Therefore, the observed differences may only reflect patient-specific local differences in the distance to criticality and cannot be attributed to the epileptogenicity of the hemisphere based on our data. One potential explanation for the lack of systematic difference in our data is that the epileptic focus might only consist in a small, localized population of neurons that was not always captured by the recording electrodes. This hypothesis is supported by evidence from epileptic seizures indicating a core territory of recruited neurons that is surrounded by a so-called ictal penumbra [51]. On the other hand, a variety of studies suggest that epilepsy is a network disease, and pre-seizure changes have been observed in neurons far away from the epileptic focus [29, 49]. Thus, it is conceivable that the entire brain operates in a different dynamical regime, which would explain the lack of a systematic difference between the hemispheres. Investigating this hypothesis would ideally require single neuron

data from healthy humans, which are, however, practically not accessible with existing recording techniques.

Interestingly, we found that the branching parameter in the hippocampus of the ipsilateral hemisphere was smaller than in the contralateral hippocampus. This contrasts our initial hypothesis of a reduced distance to criticality in the ipsilateral hemisphere. A potential explanation for the increased distance to criticality could be the destruction of neuronal tissue (e.g. mesial temporal sclerosis), which is frequently observed in MTL epilepsy [52]. Neuronal cell loss can result in a more segregated network, reducing the effective coupling between neurons and therefore the branching parameter. Another potential reason are compensatory mechanisms that reduce the coupling strength between neurons because of repeated excessive activity in that area (e.g. via homeostatic plasticity).

We found no evidence for systematic transitions from the sub- to the supercritical regime before seizure onset. As all our recordings have been obtained during seizure-free intervals (reference or pre-seizure), this finding does not contradict the experimental evidence that epileptiform activity and seizures proper might reflect a transition to the supercritical regime [26, 27]. In the literature, the reported time window of putative pre-seizure changes varies considerably from seconds to minutes and hours [30, 35, 49, 53, 54]. Even on longer time scales, recent studies suggest fluctuations in seizure susceptibility, including circadian, multidien and multimonth rhythms [55, 56]. Thus potentially, the transition to supercriticality happens on a time scale that is not detectable in the analyzed 10-minute interval. The transition could either be so rapid that our time-resolved analysis cannot capture it (faster than the 80 seconds analysis window). Or the transition could occur on a much slower time scale, such that changes during the last 10 minutes are too small to be detectable. Once the system is sufficiently close to criticality, it could spontaneously be driven into a seizure by noise. This hypothesis is supported by a recent study that found signatures of critical slowing on a timescale of hours to days before seizure onset in human iEEG [35]. A former study investigating putative pre-seizure periods of 30 minutes to 4 hours, on the other hand, did not find such signatures in human iEEG [37]. A recent study on ECoG data of human patients with focal epilepsy found clear systematic trends in signatures of critical slowing during the last 30 min before seizure onset, however these trends were not consistent across patients (increase in the signal's autocorrelation coefficient (ACC) in 4/12 patients, decrease in 4/12 and no significant change in the ACC in 4/12 patients) [30]. High frequency activity in hippocampal slices of rats, on the other hand, showed a consistent increase in the ACC as well as other signatures of critical slowing prior to seizures [30]. Thus it remains to be investigated precisely, under which conditions (e.g. types of epilepsy, types of recordings, considered time window) the framework of criticality can serve to predict an impending epileptic seizure.

Our analyses might miss signatures of a transition to supercritical dynamics because of not capturing the recruited set of neurons, or because single unit activity presents too fine a spatial scale. Single neuron activity before and during an epileptic seizure was shown to be highly heterogeneous, with some neurons increasing their firing rates, others decreasing, and a considerable fraction not changing at all [49]. In fact, evidence from spatially extended spike recordings from epilepsy patients suggests that there is a sharp boundary between areas with increased, hypersynchronous spiking and adjacent areas with low-level, unstructured activity during focal seizures [51]. This implies that recruitment of neurons to seizures occurs only in small areas and that single unit recordings potentially do not capture the recruited neurons [51]. In line with this idea, the study that found signatures of critical slowing during seizure *termination* found these signatures only on the more coarse recording levels (EEG, ECoG, LFP), not in multi unit activity [34]. Therefore, the sparsely recorded single unit activity may not be the ideal type of signal to identify seizure onset dynamics.

Our estimation of the distance to criticality relied on the branching process approximation. Mathematically, the branching process represents a generic model of how activity propagates in a network, but clearly, it is quite simplistic, and does not account for all the biological complexity in the brain. In particular, we do not distinguish excitatory and inhibitory connections. Instead, we use the branching parameter $m$ to describe the *effective* spreading of activity in the network. Any alteration in $m$ may thus reflect one or a combination of mechanisms, like altered synaptic strength, excitability, or excitation-inhibition ratio. Despite this simplification, a previous study has shown that the branching process model does indeed reproduce statistical properties of networks with inhibition, if one assumes that the excitatory and inhibitory contributions can be described by an *effective* excitation [12].

The branching process formalism has proven powerful, because in contrast to classical approaches to estimate criticality, it is (i) invariant under subsampling, returning a reliable estimate of activity spread and stability even if only a tiny fraction of neurons is sampled [5, 39, 42, 57]; (ii) it returns a precise, quantitative estimate of the distance to criticality, which enables comparison across studies, recording conditions, brain areas, and species; and (iii) it requires comparably little data, thereby enabling our time-resolved analysis. Importantly, one finds a good match between the branching process and many experimentally observed features of cortical dynamics, like spectra, Fano factors, inter-spike interval distributions, and a clear exponential decay of the autocorrelation function [6, 12, 47, 58]. Together, these aspects clearly support the validity of the branching process approximation, making it a powerful tool to assess the stability of network dynamics.

In the literature, a variety of alternative mechanisms for seizure generation have been proposed. While we focused on criticality using the framework of a branching process, several other models describe seizure generation by a critical transition (bifurcation), but with different order and control parameters [31, 32, 35, 59]. While the specific dynamics of these models differ, they all involve a deterministic change in a control parameter that could, in principle, be detectable. In fact, it is expected that all mechanisms, in which the transition to seizures involves a second order phase transition, will show signatures of critical slowing [25]. As the branching parameter $m$ is an indicator for critical slowing, it should increase if cortical dynamics approach a second order phase transition, independent of the specific underlying path. Thus, our analysis is not restricted to a branching process model, but measures the proximity of cortical dynamics to a second order phase transition.

On the other hand, there are models that do not describe seizures via bifurcations, but with multi-stable models, in which seizure onset is not driven by deterministic parameter changes, but by random fluctuations [32, 60]. In these models, seizures would be inherently unpredictable, as random fluctuations in the dynamics drive the system into the seizure state. Such a model could, in principle, explain our negative finding. However, there is accumulating evidence that seizures are, to a certain extent, predictable, which indicates that seizures are likely not only driven by noise [35, 61, 62].

Finally, a combination of mechanisms is possible: deterministic changes in the stability could lead to an increased seizure risk, accompanied by random fluctuations that drive the system into the seizure [63]. In this case, it is conceivable that the deterministic changes happened on a timescale longer than detectable in our 10-minute analysis, and entering the seizure was then driven by random fluctuations.

In addition to the variety of proposed seizure generating mechanisms, there is strong evidence that seizure onset, progression, and termination can follow different dynamical pathways [33, 56, 63]. Such differences can exist not only between patients, but also within individual patients. This renders finding a common mechanism or warning signal inherently

difficult and may provide a further explanation of why we find no consistent effect within and across patients.

We would like to conclude with a general remark on the approach of this study. We started off with two clear hypotheses about how changes in the distance to criticality might be related to seizure dynamics (Fig 1b and 1c). With the intracranial spike recordings, we could test these hypotheses, but we found no evidence for either of them. Faced with these negative results, one can either revise the hypothesis, investigate a different set of parameters or observables, e.g. by switching from a branching process instability to a framework of metastability or oscillation, and eventually one might find a hypothesis for which the data provides significant results. However, this approach can easily lead to false-positive reports. Therefore, we decided to stay with the original hypotheses, which had been formulated in all detail in a grant proposal, and then report the negative results. We believe that this is scientifically the most rigorous path to follow, though not necessarily the most rewarding. As a consequence, we have to report negative results on these specific hypotheses. However, as pointed out above, it does not exclude that other perspectives on critical phenomena might still play a role in seizure dynamics.

In summary, we found no evidence for a transition towards supercritical dynamics in pre-seizure single neuron activity of human cortex. This finding is in line with previous studies finding no warning signals of a critical transition prior to epileptic seizures [36, 37], but stands in contrast with a recent study demonstrating the effectiveness of signatures of critical slowing for seizure prediction [35]. It remains to be investigated precisely whether single-unit recordings, despite providing the most direct information on neuronal activity, cannot resolve pre-seizure changes in criticality. Furthermore, it is conceivable that seizure activity proper is indeed supercritical, with dynamics crossing the critical thresholds only seconds before seizure onset. Alternatively, seizure generation might be a qualitatively different process that cannot be captured by a linear stability parameter $m$.

## Materials and methods

### Ethics statement

All patients had given written informed consent to participate in this study, which was approved by the Medical Institutional Review Board in Bonn.

### Acquisition and pre-processing of intracranial recordings

We analyzed intracranial recordings from $n = 20$ patients with medically intractable focal epilepsy. The data was recorded at the Department of Epileptology at the University of Bonn Medical Center. For pre-surgical evaluation, patients were implanted with depth electrodes in different regions of the medial temporal lobe, including hippocampus (H), amygdala (A), parahippocampal cortex (PHC) and entorhinal cortex (EC). The location of microwires was verified using post-implantational CT scans co-registered to pre-implantational MRI scans. All wire bundles were confirmed to be located in the designated target regions in each patient. Each electrode contained 8 microwires, with which single-unit recordings could be performed. Recordings were performed continuously for pre-surgical monitoring for a typical duration of 7-14 days. Data was sampled at 32 kHz and filtered between 0.1 and 9000 Hz. Spike sorting was performed using the Combinato package [64], using the standard parameters proposed by the authors. Sorted units were classified manually as single units, multi-units, or artifacts, using the Combinato GUI [64]. The main classification criterion for putative units was the

signal's shape, but also the amplitude and distribution of inter-spike-intervals, which allows excluding artifacts due to e.g. supply voltage. For further analysis, we only used spikes of identified single units and excluded artifacts and multi-units. Thereby, we minimized the number of artifacts, which can potentially impact subsequent analyses (multi-units typically contain more artifacts than identified single units).

For each patient, we analyzed one 10-minute reference recording, obtained in a seizure-free interval after the surgery, as well as several pre-ictal recordings, spanning 10 minutes prior to seizure onset. Only clinical seizures were included in the analysis. Pre-ictal recordings end at seizure onset, which was determined by two board-certified EEG readers. Patients in which the epileptic focus could not clearly be assigned to one hemisphere were excluded from the analysis. 17 out of the 20 patients performed the surgery, and 10 patients were seizure-free afterwards (Engel scale 1a). S1 Table summarizes the analyzed patients and recordings. In total, we started with 20 patients and 116 recording periods (20 reference, 96 pre-seizure). After spike-sorting and excluding recordings that did not return any single unit, 107 recording periods (20 reference, 87 pre-seizure) remained. For each of the recording periods, we obtained data from multiple subregions, which included hippocampus, amygdala, parahippocampal cortex and entorhinal cortex, but did not always cover all subregions in each patient. For the number of recordings in each hemisphere and subregion, see S1 Text.

## Branching process approximation

We use the branching process as a minimal model for spike propagation in the brain. The branching process is a stochastic model describing the number of active neurons $A_t$ in discrete time bins of length $\Delta t$. Each active unit $i$ at time $t$ activates a random number $Y_{t,i}$ of units in the subsequent time step. In addition, there is an external input $h_t$ into the system, accounting for input from sensory modalities, from other brain areas, or spontaneous activation of individual neurons. The total number of active neurons is then given by

$$A_{t+1} = \sum_{i=1}^{A_t} Y_{t,i} + h_t. \tag{1}$$

Taking the conditional expectation value yields the autoregressive representation

$$\langle A_{t+1} | A_t = j \rangle = mj + h, \tag{2}$$

where $m = \langle Y_{t,i} \rangle$ is the branching parameter, and $h = \langle h_t \rangle$ is the average input.

A branching process is stable for $m < 1$ (subcritical regime) and unstable for $m > 1$ (supercritical regime). The critical point ($m = 1$) separates the two regimes and marks the critical point of a second-order phase transition. For more details, see [6, 65].

## Definition of the activity $A_t$

The activity $A_t$ is defined as the number of active neurons in discrete time bins $\Delta t$. Implanted depth electrodes can, however, only record a tiny fraction of all neurons, and hence one only observes a subset of the activity $A_t$. Such spatial subsampling can lead to strong biases in inferred system properties [5, 11, 18, 42, 57]. However, for estimating the branching parameter $m$, we have developed a method that overcomes the systematic bias [39, 65]. In brief, it shows that the autocorrelation strength, which is central when inferring $m$, is biased by a factor $B$; however, the bias factor $B$ can be partialled out, so that we can obtain an unbiased estimate.

In the following, we thus also denote the *sampled* activity by $A_t$. It is defined as the number of *sampled* active neurons at time $t$. To obtain $A_t$ from recorded spike times, all spikes recorded

in one brain area are pooled and binned to $\Delta t = 4$ ms time bins. The time step $\Delta t$ was chosen to reflect the propagation time of spikes between neurons.

## Definition of the autocorrelation

To estimate $m$, the autocorrelation function $C(k)$ at time lags $k$ has to be estimated from the recorded activity $A_t$:

$$
\begin{aligned}
C(k) \quad &= \frac{\text{Cov}[A_t, A_{t+k}]}{\text{Var}[A_t]} \\
&= \frac{\sum_{t=1}^{T-k} (A_t - \bar{A}_t)(A_{t+k} - \bar{A}_{t+k})}{\sum_{t=1}^{T-k} (A_t - \bar{A}_t)^2},
\end{aligned}
\tag{3}
$$

where $\bar{A}_t$ and $\bar{A}_{t+k}$ denote the mean activity of the original and the delayed time series, respectively, and $T$ the duration of the recording. This definition of the autocorrelation function $C(k)$ is equivalent to the standard definition of the Pearson correlation coefficient $\rho_{A_t, A_{t+k}} = \frac{\text{Cov}[A_t, A_{t+k}]}{\sigma_{A_t} \sigma_{A_{t+k}}}$, with standard deviations $\sigma_{A_t}$, $\sigma_{A_{t+k}}$, as long as $\{A_t\}_{t=1}^{T}$ is a stationary process and thereby $\sigma_{A_t} = \sigma_{A_{t+k}}$. If the activity $A_t$ is consistent with a stationary process with autoregressive representation (PAR), $C(k)$ decays exponentially [39].

## Estimation of the distance to criticality

We used our open source toolbox [43], which implements the Multistep-Regression (MR) estimator to infer the branching parameter $m$ from spike recordings [39]. The estimator is invariant under subsampling, i.e. it yields consistent estimates for the branching parameter of the whole system even if only a small fraction of all neurons is recorded.

For a stationary branching process, it can be shown that the autocorrelation of $A_t$ follows $C(k) \propto m^k$. The branching parameter $m$ can therefore be estimated by fitting an exponential decay

$$
\begin{aligned}
f(k) \quad &= Bm^k + D \\
&= B \exp\left(-k\Delta t / \tau\right) + D
\end{aligned}
\tag{4}
$$

to the measured autocorrelation $C(k)$. In the last term of 4, we rewrote the autocorrelation function in terms of the intrinsic timescale $\tau = -\Delta t / \log(m)$ [39, 58]. The additional offset $D$ in the fit function $f(k)$ accounts for contributions with long timescales that do not decay substantially within the recording time, and it compensates for small non-stationarities in $A_t$ [58]. The factor $B$ is the bias factor of the autocorrelation and depends on the subsampling.

In short, given the activity $A_t$, estimation of $m$ is performed in two steps [39]:

1. Compute the autocorrelation function $C(k)$ for different time delays $k$ (Eq 3).

2. Fit an exponential decay $f(k) = B \exp\left(-k\Delta t / \hat{\tau}\right) + D$ to the autocorrelation function to obtain an estimate for the intrinsic timescale $\hat{\tau}$ and the branching parameter $\hat{m} = \exp\left(-\Delta t / \hat{\tau}\right)$.

All analyses were performed using the python toolbox of the MR estimator [43]. We used time delays $k \in [4 \text{ ms}, 1600 \text{ ms}]$, which is on the order of several autocorrelation times of our data.

Confidence intervals of estimates for single recordings were obtained via a block bootstrap procedure: recordings were divided into segments of 20 s length and estimation was performed on random subsets of segments (see *stationarymean* method in [43]).

## Exclusion criteria of MR estimation

For reliable estimation of the branching parameter $m$, the data must be consistent with a stationary PAR which implies that the autocorrelation $C(k)$ must be consistent with an exponential decay. Furthermore, reliable estimation requires a minimum number of recorded spikes, because the variance in estimates increases with decreasing number of non-zero activity entries [39]. To ensure that a given time series fulfills the requirements for reliable estimation, we implemented two consistency checks:

1. Number of non-zero time bins must be at least $n_{A_t \neq 0} = 1000$.

2. The exponential decay must fit the autocorrelation better than a simple linear function. (Otherwise, the data cannot be considered consistent with a stationary PAR).

If a recording was not consistent with either of these requirements, it was excluded from the analysis. Note that at critiality, the intrinsic timescale $\hat{\tau}$ is expected to diverge. The autocorrelation function of dynamics very close-to-critical on a limited number of time steps can therefore be mistaken with a linear decay. We nevertheless implemented the second exclusion criterion, as it captured erroneous fits due to non-stationarities in the recording or high signal-to-noise ratios in our dataset. In general, however, the use of the second criterion must be assessed carefully for any given dataset, to ensure that it does not introduce a bias towards subcritical dynamics.

## Comparison of ipsilateral and contralateral hemisphere

For each recording, the spikes from the hemisphere containing the epileptic focus were combined and binned to a single activity time series $A_t^{\text{ipsi}}$. Equally, all recorded spikes from the contralateral hemisphere were binned to $A_t^{\text{contra}}$. Estimated branching parameters $\hat{m}$ of ipsilateral and contralateral hemisphere were compared by performing a mixed effects ANOVA, taking into account the patientID as a random effect. As the distribution of $\hat{m}$ was not consistent with a normal distribution, we performed the mixed effects ANOVA on the logarithmic distance to criticality $\hat{\epsilon} = \log(1 - \hat{m})$. Specifically, we used the python interface *pymer4* [66] to the R-function *lmer* [67], with model specification $\log(1 - \hat{m}) \sim \text{focus} + (1|\text{patientID})$. As differences in between patients are possible, we additionally compared ipsilateral and contralateral hemisphere separately within each of the patients.

The same approach was applied individually to each subregion of the MTL. In this case, spikes were binned separately for each subregion in each hemisphere. To disentangle potential effects of patient-ID, subregion and location of the epileptic focus, we conducted a mixed effects ANOVA, where patient-ID was modeled as a random effect. As above, we used the python interface *pymer4* [66] to the R-function *lmer* [67], with model specification $\log(1 - \hat{m}) \sim \text{focus} \cdot \text{subregion} + (1|\text{patientID})$. In addition, we performed post-hoc pairwise comparisons of $\hat{m}$ of ipsilateral versus contralateral hemisphere in all subregions using the model specification $\log(1 - \hat{m}) \sim \text{focus} + (1|\text{patientID})$.

To test whether significant differences are only present in patients with successful surgery, the comparisons of ipsilateral and contralateral hemisphere were conducted again considering only patients with ideal surgery outcome (Engel scale 1a). Furthermore, to make sure that temporal differences in the distance to criticality (e.g. due to circadian rhythm or different level of

medication) do not overshadow an effect, the analyses were conducted again only taking into account *paired* recordings, i.e. recordings obtained from the same patients during the same period of time.

### Time-resolved estimation before seizure onset

To analyze changes in the distance to criticality prior to seizure onset, we applied the MR estimator with a sliding window approach. The crucial parameter to choose is the window size $L_w$, which represents a trade-off between sufficiently short windows for high temporal resolution, and sufficiently long windows that provide enough data for a consistent estimation. Based on the average activity and the average number of non-zero activity entries in our data, we chose a window size of $L_w = 80$ s (see S4 and S5 Figs). We used overlapping windows with a fixed window-step $L_{step} = 2$ s. As a more coarse measure of pre-seizure changes, we splitted the pre-seizure recordings into two parts and estimated $\hat{m}$ separately for both parts. Estimates of the first and the second part were then compared using the Wilcoxon signed rank test, where parts of the same recording were considered pairs. Similar to the ipsilateral/contralateral analysis, we performed the statistical test on the logarithmic distance to criticality $\hat{\epsilon} = \log(1 - \hat{m})$. As an additional sanity check, we furthermore ran a mixed effect ANOVA, accounting for patientID as a random effect (model specification $\log(1 - \hat{m}) \sim$ partNo + (1|patientID) for each brain area. The results were consistent across the different testing procedures.

To assess whether the variability in the distance to criticality changes before seizure onset, we determined the variability in the time-resolved estimates of $m$ in the first and the second part of the recordings. Specifically, we computed the median absolute deviation (MAD) of the time-resolved distance to criticality $\hat{\epsilon}(t)$ from the median distance to criticality in each recording part. We then compared the MADs of first and second part using the Wilcoxon signed rank test.

## Supporting information

**S1 Fig. Autocorrelation functions (ACF) of single unit activity in human medial temporal lobe for two example patients.** ACFs were widely consistent with exponential decays with offset (lines show the fitted exponential $f(k) = Bm^k + D$, where $m$ is the estimated branching parameter). The upper row of each patient corresponds to activity in the ipsilateral hemisphere, the lower row to the contralateral hemisphere. The different plots in each row show different recordings, both reference (gray) and pre-seizure recordings (blue). (TIFF)

**S2 Fig. Patient-wise comparison of the branching parameter $m$ between the hemisphere containing the seizure onset zone and the contralateral hemisphere for both pre-seizure recordings (red) and reference recordings (orange).** While there was no consistent difference between ipsilateral and contralateral hemispheres *across* patients, *within* some of the patients there was a consistent trend in either direction (p-values of two-sided Mann Whitney-U test). (TIFF)

**S3 Fig. Comparison of ipsilateral versus contralateral hemisphere using only *paired* data, i.e. pairs of data that were recorded simultaneously in the ipsilateral and the contralateral hemisphere. a** The branching parameter $\hat{m}$ across recordings and patients shows no significant difference between ipsilateral and contralateral hemisphere. **b** Comparison of the branching parameter $\hat{m}$ in ipsilateral and contralateral hemisphere for different subregions of the MTL, (hippocampus (H), amygdala (A), parahippocampal cortex (PHC) and entorhinal cortex (EC)). The hippocampus shows a just significant difference between ipsilateral and

contralateral hemisphere, with a smaller $m$ in the ipsilateral hemisphere. The other subregions show no significant difference ($p < 0.05/4$ required after Bonferroni correction).
(TIFF)

**S4 Fig. Average population firing rate $R = \langle A_t \rangle_t / \Delta t$, and number non-zero activity entries $n_{A_t \neq 0}$, for our dataset of pre-seizure recordings.** Averages $\langle A_t \rangle_t$ are computed over all time steps of the respective recording. Across recordings, the median population firing rate is $q_{50} =$ 25.8 Hz and the 25% quantile is $q_{25} = 10.4$ Hz. The median fraction of non-zero activity entries is $q_{50} = 9.9\%$, i.e. on average, more than 90% of the time bins contain no spike.
(TIFF)

**S5 Fig. Choice of the window size for the time-resolved estimation of the branching parameter. a** Estimated $\hat{m}$ as a function of the window size for simulated branching processes. Error bars are 95% confidence intervals of estimates on 500 trials. Simulation parameters were adjusted to approximately match the median values of $\langle A_t \rangle$ and $n_{A_t \neq 0}$ of our data set ($m = 0.95$, $\alpha = 0.004$, $\Delta t = 4$ ms, $h = 1$). **b, c** $\hat{m}$ for different window sizes in two example recordings. Error bars are 95% confidence intervals of estimates from n = 1100 partially over-lapping segments of the recording (window step size of of 400 ms). The variance of estimates increases with decreasing window size. For the results shown in Fig 2c and S6 Fig, we chose a window size of 80 seconds (black arrow), representing a compromise between sufficiently high temporal resolution and sufficiently low variance.
(TIFF)

**S6 Fig. Time-resolved estimation of $\hat{m}$ within the last 10 min prior to seizure onset for all patients and recordings.** Each trace corresponds to one pre-seizure period, shown both in the ipsilateral and contralateral hemisphere. Seizure onset of each trace was at $t = 600$ s and estimated $\hat{m}$ was assigned to the middle of the respective window (window size 80 s). Recordings, in which more than 5% of segments were not consistent with the requirements for MR estimation were excluded. Note that activity enters the supercritical regime in a few segments of patient 18, but otherwise remains in the subcritical regime across patients and recordings.
(TIFF)

**S7 Fig. Comparison of ipsilateral versus contralateral hemisphere using only recordings from the 10 patients with 1a outcome after surgery. a** The branching parameter $\hat{m}$ across recordings and patients shows no significant difference between ipsilateral and contralateral hemisphere. **b** Comparison of the branching parameter $\hat{m}$ in ipsilateral and contralateral hemisphere for different subregions of the MTL, (hippocampus (H), amygdala (A), parahippo-campal cortex (PHC) and entorhinal cortex (EC)). None of the regions shows a significant difference between ipsilateral and contralateral hemisphere ($p < 0.05/4$ required after Bonferroni correction).
(TIFF)

**S8 Fig. Analysis of pre-seizure changes using only recordings from the 10 patients with 1a outcome after surgery. a** No significant difference in the distance to criticality between the last 5 min prior to seizure onset and the previous 5 min—neither in the MTL containing the epileptic focus, nor in the contralateral one (p-values of Wilcoxon signed rank test). **b** Separate analysis of the individual subregions of MTL in the hemisphere containing the epileptic focus. Due to the small number of rescordings, we report the individual estimates without perform-ing statistical tests.
(TIFF)

**S1 Text. List of recording numbers that were excluded based on the exclusion criteria for MR estimation.**
(PDF)

**S1 Table. Intracranial recordings from epilepsy patients that were analyzed in terms of the distance to criticality.** Recordings span both hemispheres (left and right) and different subregions of MTL, including hippocampus (H), amygdala (A), parahippocampal cortex (PHC) and entorhinal cortex (EC). Individual recordings of the same patient can span different subsets of the listed brain areas. Numbers of recordings after spike sorting, before applying exclusion criteria of MR estimation. The surgery outcomes were evaluated according to the Engel scale. No entry means that no surgery has been performed. Dataset provided by the Department of Epileptology in Bonn.
(PDF)

## Author Contributions

**Conceptualization:** Annika Hagemann, Jens Wilting, Viola Priesemann.

**Data curation:** Annika Hagemann, Bita Samimizad, Florian Mormann.

**Formal analysis:** Annika Hagemann.

**Investigation:** Annika Hagemann, Viola Priesemann.

**Methodology:** Annika Hagemann, Viola Priesemann.

**Supervision:** Florian Mormann, Viola Priesemann.

**Visualization:** Annika Hagemann.

**Writing – original draft:** Annika Hagemann, Viola Priesemann.

**Writing – review & editing:** Annika Hagemann, Jens Wilting, Bita Samimizad, Florian Mormann, Viola Priesemann.

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
