## [Decision Letter · Decision Letter 0]

20 May 2020

Dear Dr. Priesemann,

Thank you very much for submitting your manuscript "No evidence that epilepsy impacts criticality in pre-seizure single-neuron activity of human cortex" for consideration at PLOS Computational Biology.

As with all papers reviewed by the journal, your manuscript was reviewed by members of the editorial board and by several independent reviewers. In light of the reviews (below this email), we would like to invite the resubmission of a significantly-revised version that takes into account the reviewers' comments.

We cannot make any decision about publication until we have seen the revised manuscript and your response to the reviewers' comments. Your revised manuscript is also likely to be sent to reviewers for further evaluation. Please note that two reviewers commented on data availability which must addressed. The PLoS guidelines for this can be found here: https://journals.plos.org/ploscompbiol/s/data-availability

Sincerely,

Peter Neal Taylor

Associate Editor

PLOS Computational Biology

Samuel Gershman

Deputy Editor

PLOS Computational Biology

Reviewer's Responses to Questions

**Comments to the Authors:**

Reviewer #1: Hagemann et al. present in their paper an analysis of spatial and temporal changes in “criticality” of groups of single-neuron activity in the human brain, but claim that these changes are not related to the brain’s proximity to epileptic activity (spatially or temporally). I have some comments which I hope will help the authors improve their paper.

I have two main issues with this paper, which may simply be a matter of clarification, but definitely have implications for the interpretation of the paper by the wider scientific community.

1) As far as I can see, you do not distinguish between spikes from excitatory and inhibitory neurons. Your Fig. 1a only makes sense if all your spikes are excitatory. If you really do no distinguish the two types of spikes in the data, then I do not think you can claim the interpretation on the branching parameter as you do. M=1 is not necessarily the critical point. Please either clarify and/or provide simulations of networks of excitatory and inhibitory neurons to show the interpretation of the branching parameter.

2) I find your definition of SOZ very vague and misleading. As far as I understand it, you assign spikes as SOZ if they come from the hemisphere of the seizure focus (are these patients all MTLE patients?). This is a gross oversimplification. You could simply give it a different name to “SOZ” (which has a clear clinical definition in epilepsy), but there is still a fundamental interpretation problem: all your spikes could have come from a region that is unrelated to the seizure-generating part of the brain. The only gold standard for “epileptogenicity” is by confirming it through epilepsy surgery and showing that the patient is seizure free afterwards. The clinically-defined seizure onset zone is a working approximation to the gold standard. You need to be very clear in your paper about these concepts. Please rename your SOZ, nSOZ; provide additional analysis of spikes from the confirmed clinically-defined SOZ vs. other spikes. If surgery location and outcome is known, please provide additional analysis on spikes that come from surgically removed areas in patients that are seizure-free after surgery.

Furthermore, I have some issues with the technical and statistical aspects of this work. In no particular order:

-The spike sorting is not sufficiently described, what types of electrodes were used, I presume micro-wires? How was it determined where the microwire is located exactly for the SOZ analysis and also the brain area analysis? How does the number of microwires used for each brain area (I presume some contacts needed to be removed due to noise) affect the m^ estimations?

-You claim m^ is “unbiased by undersampling”: do you expect m^ to represent a local neural population? A brain area? A whole hemisphere? This will presumably depend on how much tissue the microwires are sampling?

-How much do spike sorting parameters affect m^?

-In the temporal analysis, is it really fair to combine all At to represent one hemisphere? Given that some brain regions were missing in some patients I’d like to see a sensitivity analysis of m^ estimates for a hemisphere with and without e.g. one (two) brain region(s).

-I would be interested in seeing equivalent results for Fig 1d&f & Fig 2b,c but using a hierarchical model/analysis where subject is modelled as a random effect. Systematic differences between subjects is possible, especially given that m=1 may not represent a universal criticality threshold depending on sampling.

-What type of seizures were included in this analysis – subclinical & clinical seizures?

Reviewer #2: In the submitted manuscript „No evidence that epilepsy impacts criticality in pre-seizure single-neuron activity of human cortex.“, Hagemann et al. analyzed single-unit recordings obtained in patients with epilepsy to test whether 1) epileptogenic tissue operates in a highly unstable regime close to the critical point of bifurcation, and 2) seizures are preceded by progressive changes in the bain stability. To estimate the stability/instability, they used an estimator that they developed and successfully applied to various datasets, including spike train data. The method estimates the branching parameter that reflects the spreading of neuronal activity.

Firstly, they determined the stability inside the SOZ zone and outside the SOZ. They did not find any differences between these regions. Current and previous findings suggest that the human brain operates physiologically close to the critical point and that epileptogenic tissue does not display higher instability (decreased stability) than the tissue outside SOZ. Secondly, the authors did not reveal any consistent changes in cellular activity that would be indicative of increasing instability with approaching seizures.

The article is focused, and the results are worth publishing. It brings an important piece of information about the cellular dynamics involved in seizure genesis and the dynamical properties of epileptogenic tissue. The discussion is written quite well. The authors critically discuss the weaknesses of a single cell approach, and they critically evaluate obtained results with the existing literature.

The manuscript is interesting and evokes many questions. Some of them should be addressed in the revised version of the article.

1. The authors should justify in greater detail the reasons for the selection of the branching estimator to determine the stability/instability. The methods estimates activity spread. Did the authors assume that the epileptogenic tissue would be characterized by a higher probability of cell recruitment or stronger connectivity between neurons? Does the hypothesis suggest that with approaching seizure, cellular recruitment will increase? Some studies suggest that spatial expansion marks the system’s decreasing resilience and distance from the bifurcation (Scheffer at al, Nature, 2009)

2. The results of current and previous studies claim that a healthy human brain operates in a highly unstable regime. Such a generalization would suggest that even a healthy brain is prone to seizure genesis?

3. How is the level of estimated stability influenced by circadian rhythm, sleep, sleep stages, or levels of vigilance?

4. The authors analyzed data from patients who underwent presurgical evaluation. Often, the antiepileptic medication is tapered at the beginning of the intracranial recording. At its end, the medication is introduced again. Is it plausible that drug withdrawal increases instability (Zijlmans et al., Neurology, 2009)? In such a scenario, the pre-ictal changes may be absent because the brain dynamics is very close to the bifurcation. The distance to the bifurcation would be minimal, and the transition to seizure very fast (basically unobservable).

5. The authors did not observe a drift in instability ahead of seizures. Apart from analyzing a trend, the authors could explore the fluctuations in the branching parameter in advance of seizures

6. In this or future studies, the authors could use antiepileptic drug withdrawal to evaluate the estimator. Or vice versa, they could assess the effect of antiepileptic drugs on brain stability.

7. Seizure onset, progression, and termination can follow a different dynamical pathway between patients (Jirsa et al., Brain, 2014; Saggio et al., J. Math. Neurosci., 2017 ). Even in individual patients, multiple seizure types can exist, each with a distinct seizure onset and dynamical profile. The authors should discuss their results from this perspective.

8. Previous studies have already proposed and demonstrated various pathways to seizure. The seminal work of Lopes da Silva should be cited (Lopes da Silva et al., Epilepsia (Suppl 12), 2003). Lopes da Silva and his colleagues proposed three possible trajectories to seizure onset from 1) stable state, 2) unstable state, and 3) via a slow drift towards the bifurcation. The work demonstrating high instability in photosensitive epilepsy should also be discussed and cited.

9. The pre-ictal changes in the stability and resilience were shown by Chang et al., Nat Neurosci, 2018. The results on the cellular dynamics should be compared with the observations in this study.

10. Recent studies demonstrate the existence of long-term fluctuations in seizure probability. Multiple rhythms can co-exist (circadian, multidien, multimonth) in each patient (Baud et al., Nat Commun, 2018; Karoly et al. Lancet Neurol. 2018). It is plausible that the process of the transition to seizure can vary between the periods of low and high seizure probability. These studies also suggest the combining the information from the multiple rhythms is highly informative about the seizure occurrence or that the slow changes in resilience can be observable at a long time scale (Chang et al., Nat Neurosci, 2018). The process of the transition to seizure occurring at the scale of minutes or hours can bring only partial information or no information at all.

11. The title should be modified because epilepsy does not play a causal role in the process.

Reviewer #3: Review is uploaded as an attachment

Reviewer #4: This manuscript examines the role of dynamical stability/instability in epilepsy, in particular by testing the hypothesis that epiletic seizures "reflect a transition to supercritical, unstable dynamics" in either of two disctinct scenarios: (a) epileptogenic areas already operate closer to instability (than nonepileptogenic areas in the same patient) during ongoing interictal periods, or (b) epileptogenic areas approach instability gradually during a preictal period leading to the seizure onset.

A branching process is used to assess stability by modeling the time evolution of population activity (number of action potentials in given time bins) computed from recorded single-units in both brain hemispheres ("focal" and "non-focal")20 patients. The branching parameter $m$ is estimated from data. When $m<1$, the system is subcritical (stable), critical for $m=1$, and supercritical for $m>1$. To examine (a) and (b) above, fixed and time varying $m$ are consider.

Estimated parameters reveal that epileptogenic areas show subcritical regimes throughout the examined interictal periods, excluding the two above hypothesized scenarios. Overall, it appears that neuronal activity in epileptogenic areas during interictal and near seizure periods are no different from near criticality, but still subcritical, regimes previously identified in various other normal or healthy state conditions.

My main concerns relate to the nature of this manuscript's contribution and its suitability to PLoSCB, and other more technical issues described in more detailed below.

(1) The manuscript does not make an important methodological or modeling contribution. Branching process and the notion of critical branching have both been examined by many groups (including the authors) in previous decades and in various contexts, including epileptic seizures. Furthermore, the estimator for the branching parameter used here has been introduced and applied to neural data by the same group in a previous publication (Nature Comm, 2018).

(2) Contrast with other related results: Despite Wilkat et al. (2019)[32] and Milanowski and Suffczynski (2016)[31], several other groups have argued (based on both theoretical and experimental results) that focal seizures arise as brain dynamics loose stability when approaching particular types of bifurcations (at seizure onset). For recent work, see for example Maturana et al. (Nature Comm, 2020) providing evidence for critical slowing prior to seizure onset based on intracranial EEG, or recent work from the Jirsa group. (This reviewer is not a co-author in those articles.) The dynamics in those cases are more complex than branching processes, especially with bifurcations into oscillations, etc. Thus, the issue seems to remain pretty open, and so the negative results presented in this study require a much more elaborate and detailed analyses, given potential issues including amount and type of data, model mismatch, etc.

(3) The authors acknowledge their negative findings could have resulted from not including data very close to seizure onsets. As acknowledged, the negative results could be explained by the possibility that loss of stability might occur in a much faster way and during a much shorter period culminating with the seizure onset.

(4) Also, a critical limitation of this manuscript is that beyond the negative results, no insights or alternative models are provided to shed new light on how brains transition into seizures. Given the results, it seems the case that brains do not transition into seizures via bifurcations (loss of stability). Which alternative generic dynamics (multi/meta-stability, or something else) and models? How data and statistical properties, etc, would differ in these different cases?

(5) While my assessement of the manuscript is based on the above, I would also like to comment on the potential issue of model mistmatch. For example, as the authors know, the equality $<y_{t,i}> = m$ for the (macroscopic) branching parameter in the conditional expectation (Eq. 1) follows from the random variables $Y_{t,i}$ being i.i.d. over neurons (i.e. sufficient homogeneity in the properties and independent activity of neurons in a network). This can be justified (via mean-field approaches, etc) for some models of large neuronal networks, but is far from being a feature of general networks or known to be case for actual neuronal networks of interest. Yes, all models are wrong, but some are usefull. The authors and others have provided some evidence branching processes capture several statistical properties of the population activity in various previous studies, although I haven't seen analysis looking at more detailed aspects such as how well probability of full sample paths are captured. Should a more refined definition of population (depending on neuronal subtypes and general cortical/neocortical network structure) matter? In some cases, it all might "average out" and not matter, but that's not clear. Furthermore, the invariance to subsampling of the branching parameter estimator should depend on how well the branching process model captures the dynamics of the target neuronal networks.

In conclusion, I think a substantially revised and extended version of the manuscript to address the above issues would be more indicated to a specialized journal in clinical neuroscience or epilepsy.</y_{t,i}>

**Have all data underlying the figures and results presented in the manuscript been provided?**

Reviewer #1: No: I find the author's approach to PLoS' Data Policy unacceptable. I would like to see a valid justification for why the spike train data (agglomerated by region at least) underlying this analysis is not published. Similarly, subsequent analysis code should also be made available.

Reproducibility of this work should be ensured as a first priority.

Reviewer #2: Yes

Reviewer #3: Yes

Reviewer #4: No: Neuronal recordings used in the analyses are not planned to be made available; only estimated models, autocorrelation functions, etc.

PLOS authors have the option to publish the peer review history of their article (what does this mean?). If published, this will include your full peer review and any attached files.

Reviewer #1: No

Reviewer #2: Yes: Premysl Jiruska

Reviewer #3: No

Reviewer #4: No
---

## [Decision Letter · Decision Letter 1]

30 Nov 2020

Dear Dr. Priesemann,

Thank you very much for submitting your manuscript "Assessing criticality in pre-seizure single-neuron activity of human epileptic cortex" for consideration at PLOS Computational Biology. As with all papers reviewed by the journal, your manuscript was reviewed by members of the editorial board and by several independent reviewers. The reviewers appreciated the attention to an important topic. Based on the reviews, we are likely to accept this manuscript for publication, providing that you modify the manuscript according to the review recommendations.

Sincerely,

Peter Neal Taylor

Associate Editor

PLOS Computational Biology

Samuel Gershman

Deputy Editor

PLOS Computational Biology

[LINK]

Reviewer's Responses to Questions

**Comments to the Authors:**

Reviewer #1: The authors have responded well to my comments.

I just have a few minor suggestions for clarity at this stage, in no particular order:

- Please rename "focal" to "ipsi", which also matches the "contra" better and is in agreement with standard epilepsy naming.

- "Our result thus suggests that there can be consistent differences in

distance to criticality between hemispheres but that the location of the epileptic focus does not predict

that difference." I think you need to be a bit more precise.

Either "the location of the epileptic focus does not predict that difference across all patients consistently" and/or "there can be consistent differences in distance to criticality between hemispheres but the direction of the effect was patient-specific".

-I think the discussion would benefit from a short sentence comment on the fact that within patients, the effect of ipsi vs contra was consistent, and what that would indicate.

Reviewer #2: I have no objections, and I recommend the manuscript to be accepted for publication in PLOS Computational Biology. The response to the referees was very exciting to read.

Reviewer #3: The authors have addressed all of my concerns. The manuscript now gives a much more balanced discussion and interpretation of results along with a more comprehensive, balanced overview of the literature. Two minor points to mention (however, I do not need to see the manuscript again):

- The discussion of Chang et al., 2018 should not only be limited to the human data results, but should, in all fairness, also mention the animal data results, which displayed clear and strong evidence of critical slowing down (Discussion section, lines 190 and following in the manuscript). Mentioning also these important results would help to provide a more balanced, also educational overview of the state of evidence to the reader.

- Reference [47] has meanwhile been published and should be cited accordingly:

[47] Antiepileptic drugs induce subcritical dynamics in human cortical networks. Christian Meisel. Proceedings of the National Academy of Sciences May 2020, 117 (20) 11118-11125; DOI: 10.1073/pnas.1911461117

Reviewer #4: The authors have addressed my concerns.

**Have all data underlying the figures and results presented in the manuscript been provided?**

Reviewer #1: None

Reviewer #2: Yes

Reviewer #3: Yes

Reviewer #4: Yes

PLOS authors have the option to publish the peer review history of their article (what does this mean?). If published, this will include your full peer review and any attached files.

Reviewer #1: No

Reviewer #2: **Yes: **Premysl Jiruska

Reviewer #3: **Yes: **Christian Meisel

Reviewer #4: No
---

## [Editor Report · Decision Letter 2]

4 Feb 2021

Dear Dr. Priesemann,

We are pleased to inform you that your manuscript 'Assessing criticality in pre-seizure single-neuron activity of human epileptic cortex' has been provisionally accepted for publication in PLOS Computational Biology.

Best regards,

Peter Neal Taylor

Associate Editor

PLOS Computational Biology

Samuel Gershman

Deputy Editor

PLOS Computational Biology

---

## [Editor Report · Acceptance letter]

4 Mar 2021

PCOMPBIOL-D-20-00388R2 

Assessing criticality in pre-seizure single-neuron activity of human epileptic cortex

Dear Dr Priesemann,

I am pleased to inform you that your manuscript has been formally accepted for publication in PLOS Computational Biology. Your manuscript is now with our production department and you will be notified of the publication date in due course.

With kind regards,

Alice Ellingham
